# Cannabis Use and Associated Risk Behavior Factors among High School Students in Mississippi: Youth Risk Behavior Surveillance System 2021

**DOI:** 10.3390/ijerph21081109

**Published:** 2024-08-22

**Authors:** Amal K. Mitra, Zhen Zhang, Julie A. Schroeder

**Affiliations:** 1Department of Public Health, Julia Jones Matthews School of Population and Public Health, Texas Tech University Health Sciences Center, Abilene, TX 79601, USA; 2Department of Epidemiology and Biostatistics, School of Public Health, College of Health Sciences, Jackson State University, Jackson, MS 39217, USA; zhen.zhang@jsums.edu; 3School of Social Work, College of Health Sciences, Jackson State University, Jackson, MS 39217, USA; julie.a.schroeder@jsums.edu

**Keywords:** cannabis, risk behavior, drug use, suicidality, CDC, YRBS

## Abstract

Cannabis is the most used illicit drug among youths in the United States. The objectives of this study were to identify the association between cannabis use and other risk behaviors, including suicidality, among high school students. This is a cross-sectional study using the 2021 Mississippi Youth Risk Behavior Surveillance System (YRBS). The 2021 YRBS data sets were combined for this study. The crude odds ratio (OR) and adjusted odds ratio (AOR) with a 95% confidence interval were generated using the survey packages in R to account for weights and the complex sampling design of the YRBS data. Univariate analysis identified seven risky behaviors that were significantly associated with current cannabis use, including carrying weapons on school campuses, suicidal attempts, electronic vapor use, current smoking, current drinking, sexual behaviors, and unsupervised children. In multivariable analysis, after adjusting for gender, race, students’ grades, and other risky behaviors, statistically significant variables for cannabis use included current use of electronic vapor, current smoking, current drinking, and sexual behaviors. Cannabis use is evenly burdened between males and females and between all race categories among Mississippi high school students. The identified associations seem to indicate that electronic vapor, tobacco products, and alcohol use could be the forerunners for drug use and should be treated accordingly in drug use prevention programs.

## 1. Introduction

Cannabis is one of the most widely used substances among youths and adolescents in the United States. According to the National Center for Drug Abuse Statistics, 2.08 million of 12 to 17-year-olds nationwide reported using drugs in the last month; of them, about 84% reported using cannabis [1]. The use of cannabis has been associated with short-term and long-term side effects such as increased heart rates, mood changes, difficulty with thinking and problem-solving, and impaired memory [2]. People who started smoking cannabis heavily in their teens had long-term dysfunction of the brain (notable IQ declines) and mental disorders, including anxiety, depression, and suicidal thoughts among teens [2]. One of the brain areas called the prefrontal cortex enables the brain to assess situations, make sound decisions, and keep our emotions under control. In adolescents and children, this part of the brain continues to grow into adulthood and undergoes dramatic changes during adolescence [3]. Introducing drugs during this critical period of brain development can cause profound and long-lasting effects on brain function. Studies show that drug use can exacerbate existing mental health issues, resulting in persons being two to four times more likely to develop full-blown psychiatric disorders [3,4]. Teens who use cannabis are not only more likely to experience major mental health problems (anxiety and depression), but they are also at risk of developing psychosis, if genetically predisposed. They also risk developing diagnosable substance use disorders (SUDs), which can complicate chronic health conditions and result in negative social and economic consequences [3,5].

In our recently published report, although the burden of drug use in adolescents is showing a declining trend in the United States, Mississippi’s drug use trend has been increasing in the last 20 years from 2001 to 2021 [6]. This calls for exploring the root causes of the higher trend in Mississippi.

Research indicates a link between the use of various substances and continued cannabis use. While previous studies indicate a declining trend in cannabis use over time, rates for the underlying risky behaviors are not steady. For example, cigarette smoking rates have dropped significantly from 28.3% in 1996 to 2.3% in 2022; however, the prevalence of nicotine vaping increased from 2014 to 2019 and then decreased post-2019 [7]. In 2023, the lifetime use prevalences of nicotine vaping were higher as students’ grades increased: 11.4% in 8th graders, 17.6% in 10th graders, and 23.2% in 12th graders. Adolescents who vape are at a higher risk of progressing to cannabis and alcohol use [8]. For instance, current users of electronic vapor products are 9.3 times more likely to use cannabis [9]. Alcohol use and abuse remain prevalent among adolescents and are often used with cannabis [10]. In 2023, 30.6% of 10th graders and 45.7% of 12th graders reported using alcohol in the past year, with 35% currently using it [8].

While extensive research exists on cannabis, alcohol, and other illicit substance use among adolescents nationally, our study is the first to focus specifically on Mississippi, a state with unique demographic characteristics, including higher poverty rates and a larger proportion of racial and ethnic minorities compared to the national average [11]. This study addresses a critical gap in the literature by examining the relationship between demographic factors, other risky behaviors, and substance use in Mississippi’s youth population, an area that has been understudied despite the state’s increasing trend in drug use over the past two decades.

Therefore, by utilizing the Youth Risk Behavior Survey (YRBS) data from the CDC, we offer the first in-depth analysis of how Mississippi’s distinctive socioeconomic landscape influences adolescent cannabis use, filling a crucial knowledge gap in the existing literature.

## 2. Material and Methods

### 2.1. Data Source

The Youth Risk Behavior Survey (YRBS) is a cross-sectional survey that monitors health risk behaviors in high school students. The Centers for Disease Control and Prevention (CDC) developed the YRBS for the national and state levels. As a complex sampling survey, the YRBS used a multistage probability design to ensure a nationally representative sample. The primary sampling unit (PSU) is at the county level. The secondary sampling unit (SSU) is defined at the school level. PSUs and SSUs are sampled with a probability proportional to the overall school enrollment size. The third stage is random sampling, involving one or two classrooms in each grade 9–12 of the selected schools [12,13]. The Mississippi YRBS is conducted by the Mississippi Departments of Health and Education every other year. Mississippi YRBS 2021 data were obtained from the CDC public domain [14].

### 2.2. Measurements

This study’s data point of interest was current cannabis use among Mississippi high school students. The self-reported question “During the past 30 days, how many times did you use cannabis (marijuana)?” captured six possible responses (0 times; 1 or 2 times; 3 to 9 times; 10 to 19 times; 20 to 39 times; and 40 or more times). These responses were recoded into a binary variable for analysis. We examined the associations between current cannabis use and other YRBS health risk behaviors that contributed to the leading causes of death, disability, and social problems. The YRBS monitors several categories of health risk behaviors: behaviors that contribute to unintentional injuries and violence; tobacco use; alcohol and other drug use; sexual behaviors related to unintentional pregnancy and sexually transmitted infections; and unhealthy dietary behaviors and physical inactivity. Using domain knowledge gained from a literature review of past research on the subject and using univariate selection by setting the *p*-value at 0.25 [15,16], the following factors were selected for the multivariable logistic regression model that generated the adjusted odds ratio: Carried a weapon on school property (Yes, No); Attempted suicide (Yes, No); Current use of electronic vapor products (Yes, No), including e-cigarettes, vapes, mods, e-cigs, e-hookahs, or vape pens, such as JUUL, Vuse, NJOY, Eif, Bar, or Esco Bars; Currently smoked cigarettes or cigars (Yes, No); Currently drank alcohol (Yes, No); Having sex of sex partner(s) (Never had sex, Opposite sex only, Same-sex only or both sexes); and Usually did not sleep in their parent’s or guardian’s home (Yes, No). The variable “Usually did not sleep in their parent’s or guardian’s home” was selected based on the assumption that parental presence helps to prevent drug use. In other words, we considered unsupervised children to be at a greater risk of drug use. These were all binary variables recoded from responses to self-reported questions.

The data points used in this study also included gender (male, female) and race/ethnicity (Non-Hispanic American Indian or Alaska Native, Non-Hispanic Asian, Non-Hispanic Black, Hispanic, Non-Hispanic Native Hawaiian or Other Pacific Islander, Non-Hispanic White, and Non-Hispanic Multiple Races). Due to the small Mississippi YRBS data sample size, race was recoded to have three values for analysis: White (Non-Hispanic White), Black (Non-Hispanic Black), and Other. The “Other” race category included Hispanic, Asian, American Indian/Alaskan Native, Native Hawaiian/Other Pacific Islander, and multiple races [17,18].

### 2.3. Statistical Analysis

First, a crude odds ratio (OR) with a 95% confidence interval (95% *CI*) was generated by using univariate logistic regression models between current cannabis use, each of the health risk behavior factors, and demographic variables. Then, an adjusted odds ratio (AOR) with a 95% *CI* was obtained by applying multivariable logistic regression models that account for confounding and effect modification. When the 95% *CI* contained 1 but the *p*-value was less than 0.05, we reported the difference as statistically significant [19,20].

R software version 4.4.0 was used for sample characteristic statistics, summary statistics, and logistic regression models. The CDC identifies the survey packages in R as appropriate tools capable of accounting for the complex sampling design of the YRBS data [21].

## 3. Results

### 3.1. Sample Characteristics

The Mississippi 2021 YRBS sample size was 1747. Female and male students were approximately equally represented, with the valid percent being 49.9% and 50.1% for female and male, respectively. There were 35.4% White, 49.2% Black, and 15.4% Other races. The valid percentages for students were 36.6% for 9th grade, 23.1% for 10th grade, 22.6% for 11th grade, and 17.7% for 12th grade.

Table 1 shows that the overall prevalence of current cannabis use was 13.4%. There was no significant difference in current cannabis use between female and male students (PD = −1.8, PR = 1.14, *p* = 0.32). Compared to White students, no significant difference was observed in current cannabis use for Black students (PD = 3.1, PR = 1.27, *p* = 0.09) or students in the Other race subgroup (PD = 4.5, PR = 1.39, *p* = 0.06). Compared to the 9th graders, no statistically significant difference was observed in current cannabis use for students in other grade categories.

### 3.2. Associated Factors for Cannabis Use

Table 2 presents the evaluation of the association between current cannabis use and other health risk behavior factors in terms of the crude *OR* and *AOR*.

The univariate analysis identified seven risky behaviors that were significantly associated with current cannabis use. (1) Carrying weapons on school campuses: 6.2% of students carried weapons on school properties. These students were 3.8 times more likely to be current cannabis users (*p* < 0.001) compared to those who did not. (2) Suicidal attempt: 16.2% of students attempted suicide. These students were 3.6 times more likely to be current cannabis users (*p* < 0.001) compared to those who did not. (3) Electronic vapor use: 20.9% of students currently use an electronic vapor product. These students were 24 times more likely to be current cannabis users (*p* < 0.001) compared to those who did not. (4) Current smoking: 8.6% of students smoke cigarettes or cigars. These students were 18.4 times more likely to be current cannabis users (*p* < 0.001) compared to those who did not. (5) Current drinking: 21.9% of students currently drink alcohol. These students were 11.2 times more likely to be current cannabis users (*p* < 0.001) compared to those who did not. (6) Sexual behavior: 39.1% of students had sexual contact with the opposite sex only. These students were 10.6 times more likely to be current cannabis users (*p* < 0.001) compared to students who never had sex. About 11% of students had sexual contact with the same sex only or both sexes; these students were 19 times more likely to be current cannabis users (*p* < 0.001) compared to those who never had sex. (7) Unsupervised children (children not sleeping with parents or guardians): 7.5% of students did not sleep in their parent’s or guardian’s house. These students were 4 times more likely to be current cannabis users (*p* < 0.001) compared to those who slept in their parent’s or guardian’s home.

The *AOR* in Table 2 shows the strength of the association between a risk factor and current cannabis use after adjusting for gender, race, grade, and other risk factors. The association between cannabis use and risky behaviors, such as “Carrying weapons on school properties”, “Attempted suicide”, and “Usually did not sleep in their parent’s or guardian’s home”, was no longer statistically significant after taking confounding and effect modification into the calculation. Students who currently used an electronic vapor product or smoked cigarettes or cigars and students who currently drank alcohol were still shown to more likely be current cannabis users compared to those who did not, with the *AOR* being 12.8 (*p* < 0.01), 4.4 (*p* < 0.05), and 2.8 (*p* < 0.05), respectively. Compared to students who never had sex, those who had sexual contact with the opposite sex only or those who had sexual contact the same sex only or both sexes were more likely to be current cannabis users, with the *AOR* being 2.7 (*p* < 0.05) and 3.0 (*p* = 0.05), respectively.

## 4. Discussions

In this study, the prevalence of current cannabis usage among high school students was 13.4%, which is lower than the national average of 27.8% [22]. However, this study provided new data on the prevalence of seven risky behaviors, such as carrying weapons on school property, attempted suicide, current use of electronic vapor products, current smoking, current drinking, sexual behavior, and unsupervised children (not living with parents or guardians) in Mississippi. These results are crucial for policymakers in targeting adolescents for drug prevention strategies.

Although the rate of girls using cannabis outnumbered boys in the nation (30.9% vs. 24.8% for girls and boys, respectively), there was no statistical difference based on gender in our study in Mississippi.

Cannabis usage also varied significantly based on grade level in the national sample. In contrast, there was no significant difference in the use rate based on grade level in Mississippi, although the rates increased among seniors compared to other high school students. The use prevalence rates of cannabis in the nation are disproportionately higher among seniors—about 1 in 16 high school seniors use cannabis every day [23]. About 21% of 12th graders and 10% of 10th graders reported smoking cannabis in the last 30 days in the survey [23], whereas in Mississippi, the rates are about the same between 10th graders and 12th graders. The observed differences between the national and Mississippi rates could be attributable to demographic differences between the two populations.

The associations of the seven behavioral risk factors were analyzed in our study. They played a significant role in cannabis usage in Mississippi. In the univariate analysis, all of the behavioral risk factors studied, including carrying weapons on school campus, suicidal attempts, current use of electronic vape products, current smoking of cigarettes or cigars, current drinking, sexual activities (sex with the opposite sex only or same sex or both sexes), and not staying at one’s parent’s or guardian’s home (meaning unsupervised children) were significantly associated with current cannabis use (*p* < 0.001 for all). In the multivariable logistic regression analysis, all of the factors except “Carried a weapon on school property”, attempted suicide, and not staying in the parental house remained statistically significant factors for cannabis use.

A review article examined 95 published papers that found a concurrent use pattern of alcohol and cannabis [24]. The authors analyzed the data using substitution versus complementary theories of use. Substitution theory posits that persons might use one substance in place of another due to similarities in the drug’s effect [25]. Complementary theory, on the other hand, posits that persons may use drugs in combination to enhance their outcomes [26]. Interestingly, both theories held for the concurrent use of alcohol and cannabis. The data also suggest that the coadministration of alcohol and cannabis (marijuana) results in increased impairment of brain function compared to single use of either substance [27].

In another study of 2034 college students, using both alcohol and cannabis [28] were assessed to determine whether protective behavioral strategies (PBSs) were helpful for both substances among concurrent users. In the study mentioned here, PBSs had mediator effects on alcohol and cannabis outcomes, suggesting that strategies such as PBSs seem to be an essential intervention target for alcohol/cannabis concurrent users. Although it is not entirely clear what accounts for these concurrent trends, researchers showed a possible bidirectional association between cannabis consumption and other risky behaviors, such as involvement in physical violence, among adolescents [29] and also between cannabis intake and depressive symptoms among young people [30]. Issues such as concurrent trends of alcohol and cannabis intake could also be examined using the Common Liability Theory and gateway hypothesis. Common Liability Theory highlights the underlying vulnerabilities that predispose individuals to substance use, regardless of the specific substance. The gateway hypothesis explains the progression of substance use, where initial, seemingly harmless behaviors can escalate to more dangerous forms of drug use over time [31,32]. For example, the concept of CLA leans toward a common pathway of biobehavioral disorder that invites youth’s risk behaviors. In contrast, the gateway hypothesis uses a “soft” risky behavior, such as alcohol intake, as a gateway to many more dangerous behaviors, such as using more harmful drugs. Together, these theories can help inform prevention and intervention strategies by addressing both the underlying liabilities and the specific sequences of behavior that lead to more severe substance use disorders [32].

Our study concurred with a nationally representative cohort study, Population Assessment of Tobacco and Health (PATH), in which 9828 adolescents’ data were analyzed [33]. The PATH study found a strong association between e-cigarette use and subsequent cannabis use in 1 year. In this study, those who had ever used e-cigarettes had a 2.57 times greater risk (adjusted relative risk) of cannabis use in the subsequent 12 months (95% *CI*, 2.04–3.09), after adjusting for sociodemographic characteristics [27].

Like our study, other studies have demonstrated an association between suicidality (or a suicidal attempt) and cannabis use [34,35]. However, a causal association could not be established in most studies because it was not certain which came first—whether the suicide attempt was a result of a mental disorder (such as depression) or whether depression was a result of substance use. However, in a systematic review and meta-analysis of 11 studies and 23,317 individuals, adolescent cannabis consumption was associated with an increased risk of developing depression and suicidal behavior later in life [35].

The use of cannabis is generally associated with more frequent sexual activity. In our study, sexually active children (those who had sexual contact with the opposite sex only or those who had sexual contact with the same sex only or both sexes) were more likely to be current cannabis users compared to those who had never had sex. Although this association is a new finding, a causal association between the type of sexual partner (same sex, opposite sex, or both) as a risky behavior and cannabis intake cannot be established from this study. However, some studies suggest that current cannabis users were about 8 times more likely than those who had never used cannabis to have multiple sex partners (more than two recent sexual partners) (OR 8.1, 95% *CI*: 1.94 to 33.44) [36]. On the other hand, some studies suggest that the frequency of cannabis use before partnered sex significantly correlated with increased orgasm frequency for women who experienced orgasm difficulties, meaning cannabis could have therapeutic benefits on sexual function [37].

### Limitations

Our cross-sectional study has several limitations; although several risk factors were significantly associated with cannabis use, a causal association was undermined, as mentioned earlier. Another potential limitation of this cross-sectional study is the reporting bias (either under-reporting or over-reporting) of self-reported events of cannabis use and the behavioral risk factors. The limited sample size (*n* = 1747) is an issue for the generalizability of the data. However, we included all of the data available from the YRBS. Another potential limitation of this study is that a few socioeconomic variables, such as household income and parental education, are unavailable in the YRBS data set. In addition, because of the lack of data, we failed to adjust for potential confounding effects of peer pressure or school type in this study.

However, the CDC’s YRBS data are robust as they use a multistage stratified sampling design with a large representative sample of the nation. So far, this is the only study that has reported the association of multiple demographic and behavioral risk factors with cannabis use in the high school population in Mississippi.

## 5. Conclusions

This study revealed significant associations between various behavioral risk factors, such as cigarette smoking, the use of electronic vapor products, alcohol consumption, and sexual activity, with concurrent cannabis use among high school students in Mississippi. Our study uniquely combines an analysis of youth risk behaviors, suicidality, and cannabis use in Mississippi, providing a comprehensive picture that has not been previously explored in this specific context. This study is particularly timely and relevant given the contrasting trends observed in Mississippi—where drug use has been increasing over the last 20 years—compared to the declining national trends, offering valuable insights for targeted intervention strategies. Implementing intervention strategies like PBSs and targeted risk reduction programs for at-risk groups could play a crucial role in addressing the increasing prevalence of substance abuse and its negative impacts on youth. School-based initiatives and preventive measures, such as offering academic enrichment, promoting socially competent behaviors, providing social skills training, and fostering school–family–community partnerships, are essential strategies for reducing behavioral risks among school children.

## Figures and Tables

**Table 1 ijerph-21-01109-t001:** Current cannabis use prevalence of Mississippi high school students by demographic characteristics according to Mississippi YRBS 2021.

Characteristic	Prevalence (%)	PD * (%)	PR * (95% *CI*)	*p*-Value
Total	13.4 (11.8, 15.1)	NA *	NA *	NA *
By Gender				
Female	14.3 (11.8, 16.9)	Ref *	Ref *	
Male	12.5 (10.2, 14.9)	−1.8	1.14 (0.88, 1.48)	0.32
By Race				
White	11.7 (9.3, 14.0)	Ref *	Ref *	
Black	14.8 (12.1, 17.5)	3.1	1.27 (0.96, 1.69)	0.09
Other	16.2 (11.2, 21.1)	4.5	1.39 (1.00, 1.94)	0.06
By Grade				
Grade 9	11.6 (8.6, 14.6)	Ref *	Ref *	
Grade 10	14.0 (9.5, 18.4)	2.4	1.20 (0.77, 1.87)	0.42
Grade 11	13.2 (10.2, 16.2)	1.6	1.13 (0.75, 1.70)	0.55
Grade 12	15.0 (10.8, 19.1)	3.4	1.29 (0.87, 1.90)	0.21

* PD—prevalence difference; PR—prevalence ratio; Ref—reference group; NA—not applicable.

**Table 2 ijerph-21-01109-t002:** Behavior risk factors associated with current cannabis use according to Mississippi YRBS 2021.

Risk Factors	Risk Factor Prevalence	Univariate Analysis		Multivariable Analysis	
		OR * (95% *CI*), %	*p*-Value **	AOR * (95% *CI*), %	*p*-Value **
Carried a weapon on school property					
No		Ref *		Ref *	
Yes	6.2%	3.8 (2.2, 6.4)	<0.001	0.9 (0.2, 4.5)	0.88
Attempt suicide					
No		Ref *		Ref *	
Yes	16.2%	3.6 (2.1, 6.4)	<0.001	1.2 (0.5, 2.6)	0.72
Currently used an electronic vapor product					
No		Ref *		Ref *	
Yes	20.9%	24.0 (15.8, 36.3)	<0.001	12.8 (7.9, 20.6)	**<0.001**
Currently smoked cigarettes or cigars					
No		Ref *		Ref *	
Yes	8.6%	18.4 (13.0, 26.1)	<0.001	4.4 (2.0, 10.0)	**<0.05**
Currently drank alcohol					
No		Ref *		Ref *	
Yes	21.9%	11.2 (7.8, 16.0)	<0.001	2.8 (1.6, 4.7)	**<0.05**
Sex of sex partner(s)					
Never had sex		Ref *		Ref *	
Opposite sex only	39.1%	10.6 (5.5, 20.5)	<0.001	2.7 (1.3, 5.4)	**<0.05**
Same sex only or both sex	10.8%	19.0 (8.7, 41.5)	<0.001	3.0 (1.3, 7.4)	**0.05**
Usually did not sleep in their parent’s or guardian’s home					
No		Ref *		Ref *	
Yes	7.5%	4.0 (2.8, 5.9)	<0.001	2.9 (0.8, 10.3)	0.13

* OR—odds ratio; AOR—adjusted odds ratio; Ref—reference group. ** Bold *p*-value indicates significance.

## Data Availability

Publicly archived data sets from the Centers for Disease Control and Prevention were analyzed or generated during this study. The data are available at the following link: https://www.cdc.gov/healthyyouth/data/yrbs/index.htm (accessed on 10 June 2024).

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
