# Peer review of "Cannabis Use and Associated Risk Behavior Factors among High School Students in Mississippi: Youth Risk Behavior Surveillance System 2021"

_ijerph, 2024, doi:10.3390/ijerph21081109_

Round 1

Reviewer 1 Report

Comments and Suggestions for Authors

In the attached document

Comments on the Quality of English Language

Lines 122-124: The sentence in this section should be reviewed as it is unclear. 

Reviewer 2 Report

Comments and Suggestions for Authors

Analyzing the 2021 Mississippi Youth Risk Behavior Survey data, the authors investigated the association between marijuana/cannabis use and six health risk behaviors (nicotine vaping, cigarette or cigar smoking, alcohol use, attempt suicide, sexual behavior, and sleeping in parent’s or guardian’s home. The regression results showed that cannabis use was associated with the first three outcomes and sexual behavior. The focus on Mississippi is novel given the distinctive socioeconomic landscape. However, the limited sample size (n=1747) and the lack of socioeconomic variables in the regressions (such as household income or parental education) made this study less interesting.

General:

Some paragraphs are highlighted in yellow. Why is that?

Replace marijuana with cannabis. The word marijuana, due to the historical context, is considered as pejorative and racist.

Abstract

1. Move all the survey questions, such as “currently used an electronic vapor product”, to the Methods of the manuscript.

2. Explicitly list the outcomes of this study.

Introduction

1. Line 65-66, not clear how nicotine vaping has risen since only 2023 data was mentioned. The prevalence of nicotine vaping increased from 2014-2019 and then decreased post 2019.

Are these rates (11.4%, 17.6%, 23.2%) current use or lifetime use of e-cigarettes? I think they are lifetime use prevalence.

Material and Methods

1. Line 106: “current use of vapor products” does not necessarily mean nicotine vaping, which is how the authors interpreted. Vapor products can also be used to vape cannabis. Clarify if the survey question asked about nicotine vaping or other vaping. If the question did not specify, then add this as a limitation.

2. Section 2.4 can be omitted.

3. Table 1, are there other sociodemographic characteristics that can be included? The authors mentioned in the Introduction that Mississippi is unique due to its high poverty rate but the variables here don’t capture income.

4. Line 179, the AORs in Table 2 adjusted for the other risk factors, but it’s not clear if the demographic variables in Table 1 were adjusted as well.

5. What is the prevalence of risk factors? Can be added in Table 1.

6. I understand that substance use (e-cigarette, cigarettes/cigars, alcohol), suicide attempt, and sexual behaviors are risky behaviors. But why is sleeping in parent’s or guardian’s home a risky behavior? And how are these behaviors selected? Are there more behaviors that can be added?

Discussion

1. Half of the discussion is focused on alcohol use, which is only one of the six outcomes.

2. The paragraph between line 223-236 is not very related to the findings of this study.

3. There is no discussion on the outcome of sleeping in parent’s or guardian’s home.

Comments on the Quality of English Language

NA

Round 2

Reviewer 2 Report

Comments and Suggestions for Authors

The authors have addressed all of my previous comments. 

Author Response

I made the changes as suggested.
